# Repeatability and reproducibility of a handheld quantitative G6PD diagnostic

Benedikt Ley[1]*, Ari Winasti Satyagraha[2], Mohammad Golam Kibria[3], Jillian Armstrong[4], Germana Bancone[5,6], Amy K. Bei[4], Greg Bizilj[7], Marcelo Brito[8], Xavier C. Ding[9], Gonzalo J. Domingo[7], Michael E. von Fricken[10], Gornpan Gornsawun[5], Brandon Lam[11], Didier Menard[12,13,14], Wuelton Monteiro[8], Stefano Ongarello[9], Sampa Pal[7], Lydia Visita Panggalo[2], Sunil Parikh[4], Daniel A. Pfeffer[1], Ric N. Price[1,6,15], Alessandra da Silva Orfano[4], Martina Wade[4], Mariusz Wojnarski[16], Kuntawunginn Worachet[16], Aqsa Yar[12], Mohammad Shafiul Alam[3], Rosalind E. Howes[9]

1 Global and Tropical Health Division, Menzies School of Health Research and Charles Darwin University, Darwin, Australia, 2 Eijkman Institute for Molecular Biology, Jakarta, Indonesia, 3 International Center for Diarrheal Disease Research, Bangladesh, Dhaka, Bangladesh, 4 Yale School of Public Health, Department of Epidemiology of Microbial Diseases, New Haven, Connecticut, United States of America, 5 Shoklo Malaria Research Unit, Mahidol-Oxford Tropical Medicine Research Unit, Faculty of Tropical Medicine, Mahidol University, Mae Sot, Thailand, 6 Centre for Tropical Medicine and Global Health, Nuffield Department of Clinical Medicine, University of Oxford, Oxford, United Kingdom, 7 PATH, Seattle, Washington, United States of America, 8 Fundação de Medicina Tropical Dr. Heitor Vieira Dourado, Manaus, Brazil, 9 FIND, Geneva, Switzerland, 10 George Mason University, Fairfax, Virginia, United States of America, 11 Johns Hopkins University School of Medicine, Baltimore, Maryland, United States of America, 12 Institut Pasteur, INSERM U1201, Paris, France, 13 Laboratoire de Parasitologie et Mycologie Médicale, Les Hôpitaux Universitaires de Strasbourg, Strasbourg, France, 14 Institut de Parasitologie et Pathologie Tropicale, UR7292 Dynamique des interactions hôte pathogène, Fédération de Médecine Translationnelle, Université de Strasbourg, Strasbourg, France, 15 Mahidol-Oxford Tropical Medicine Research Unit (MORU), Faculty of Tropical Medicine, Mahidol University, Bangkok, Thailand, 16 Armed Forces Research Institute of Medical Sciences, Bangkok, Thailand

* Benedikt.Ley@menzies.edu.au

**Data Availability Statement:** All relevant data are within the manuscript and its Supporting Information files.

## Abstract

### Background

The introduction of novel short course treatment regimens for the radical cure of *Plasmodium vivax* requires reliable point-of-care diagnosis that can identify glucose-6-phosphate dehydrogenase (G6PD) deficient individuals. While deficient males can be identified using a qualitative diagnostic test, the genetic make-up of females requires a quantitative measurement. SD Biosensor (Republic of Korea) has developed a handheld quantitative G6PD diagnostic (STANDARD G6PD test), that has approximately 90% accuracy in field studies for identifying individuals with intermediate or severe deficiency. The device can only be considered for routine care if precision of the assay is high.

### Methods and findings

Commercial lyophilised controls (ACS Analytics, USA) with high, intermediate, and low G6PD activities were assessed 20 times on 10 Biosensor devices and compared to spectrophotometry (Pointe Scientific, USA). Each device was then dispatched to one of 10 different

**Funding:** This study was funded by a grant from the Australian government (DFAT) to the Foundation for Innovative New Diagnostics (FIND). RNP is a Wellcome Senior Fellow in Clinical Science (200909), and GB and GG are in part funded by the Wellcome Trust (220211). The funders had no role in study design, data collection and analysis, decision to publish, or preparation of the manuscript.

**Competing interests:** The authors have declared that no competing interests exist.

laboratories with a standard set of the controls. Each control was tested 40 times at each laboratory by a single user and compared to spectrophotometry results.

When tested at one site, the mean coefficient of variation (CV) was 0.111, 0.172 and 0.260 for high, intermediate, and low controls across all devices respectively; combined G6PD Biosensor readings correlated well with spectrophotometry ($r_s = 0.859$, p<0.001). When tested in different laboratories, correlation was lower ($r_s = 0.604$, p<0.001) and G6PD activity determined by Biosensor for the low and intermediate controls overlapped. The use of lyophilised human blood samples rather than fresh blood may have affected these findings. Biosensor G6PD readings between sites did not differ significantly (p = 0.436), whereas spectrophotometry readings differed markedly between sites (p<0.001).

## Conclusions

Repeatability and inter-laboratory reproducibility of the Biosensor were good; though the device did not reliably discriminate between intermediate and low G6PD activities of the lyophilized specimens. Clinical studies are now required to assess the devices performance in practice.

### Author summary

Novel treatment regimens for the radical cure of *P. vivax* malaria are more effective than current options but require prior quantitative G6PD testing. The reference method for quantitative G6PD measurement is spectrophotometry but, due to its operational characteristics, is not suitable for routine use. Furthermore, poor inter-laboratory reproducibility of spectrophotometry has prevented quantitative global definitions of G6PD deficiency. SD Biosensor (ROK) have developed a novel handheld "Biosensor" device (G6PD STANDARD), which measures G6PD activity within two minutes and has operational characteristics suited to point of care diagnosis. Reported accuracy of the Biosensor against spectrophotometry is around 90%, but its reproducibility remains unknown. This article reports the reproducibility of the device. Standardized samples were tested in two phases, first by a single user on ten Biosensors and then by ten independent users. All users received a standardized one-hour online training. Measured G6PD activities did not differ significantly across all devices in either phase, demonstrating the capacity to provide user-independent results. If further studies under real life conditions generate comparable results, the Biosensor will allow global cut offs for G6PD deficiency to be defined and will greatly simplify the roll out of novel highly effective radical cure treatment regimens for *P. vivax* infections.

## Introduction

The 8-aminoquinolines primaquine and tafenoquine are the only drugs currently on the market with hypnozoitocidal properties, important for the clearance of *Plasmodium vivax* and *P. ovale* from the human host [1–3]. Well tolerated in the majority of recipients, 8-aminoquinolines are strong oxidants that can cause hemolysis in individuals with low activity levels of the glucose-6-phosphate dehydrogenase enzyme (G6PD), known as G6PD deficiency (G6PDd) [4, 5]. The G6PD gene is located on the X-chromosome, males are either hemizygous deficient or normal, whereas females are homozygous deficient, normal, or heterozygous for the gene.

Heterozygous females have two distinct red blood cell (RBC) populations, G6PD normal and G6PD deficient, that circulate in a ratio determined through the random process of lyonization [6]. Therefore, the G6PD activity levels of heterozygous females–and their associated hemolytic risk—is dependent on the proportion of deficient cells, those cells at greatest risk of drug induced hemolysis. Approximately 400 million people worldwide are affected by G6PDd, with allele frequencies reaching up to 35% in malaria endemic areas [7, 8]. Accordingly the WHO recommends routine G6PD testing prior to radical cure (schizontocidal and hypnozoitocidal treatment) with primaquine, whenever possible [9]. A 14-day course of primaquine is prescribed to patients with more than 30% G6PD enzyme activity, while eight weekly doses are recommended in patients with less than 30% activity [9]. These long treatment courses affect treatment adherence, and lead to lower effectiveness [10, 11]. Short course, high dose primaquine treatment regimens, as well as a single dose tafenoquine treatment regimen, are likely to improve effectiveness, however these will require more stringent criteria to protect those at risk of hemolysis [12, 13]. While qualitative G6PD diagnostics have good discriminatory power at the 30% activity threshold, they cannot discriminate patients at higher G6PD activity levels [14]; to date this can only be done by quantitative spectrophotometry [15, 16]. Not only is spectrophotometry logistically unsuitable for supporting case management of *P. vivax* patients in remote areas where most patients live, spectrophotometry has also been shown to exhibit significant variability in its measurements [17]. For example, definitions of 100% G6PD enzyme activity in U/gHb differ significantly between studies [15]. Given the current definitions of "G6PD deficient" (<30% of normal G6PD levels) and "intermediate" (30–70% or 30%-80% of normal levels), this leads to different diagnostic cut-offs between areas [18, 19]. Comparisons of G6PD activity of standardized quality control samples show significant variation between laboratories, suggesting that at least some of the variability observed in population-level G6PD readings may be due to the spectrophotometric assay itself [15]. This diagnostic variability confounds the definition of global absolute cut-offs for case management. The consequence of this assay-derived variability is that site and assay specific G6PD baseline (100% activity) levels need to be established before local deficient and intermediate thresholds can be set, adding significant complexity to the roll out of G6PD testing in *P. vivax* endemic settings [18]. G6PD levels are affected by RBC density [20], any G6PD measurement therefore needs to be normalized by an Hb reading and this may also contribute to the observed variability in spectrophotometry reading.

A hand-held quantitative G6PD diagnostic has been developed by SD Biosensor (STANDARD G6PD test, Suwon-si, ROK), hereafter referred to as the "Biosensor". The device consists of the Biosensor and a single use test strip that is inserted into the Biosensor. To generate a reading, 10µl of blood are added to a lysis buffer, 10µl of the blood buffer solution are then added to the single use test strip inserted into Biosensor. The test strip contains 5-bromo-4-chloro-3-indolyl-phosphate (BCIP) that is reduced to violet nitro blue tetrazolium (NBT) in the presence of the G6PD enzyme, the color intensity is directly proportional to G6PD activity and is measured through reflectance photometry. The Biosensor device quantifies Hb concentration using a photo-reflectance based algorithm informed by the sample's color intensity. This is measured on a separate spot to that for the G6PD activity. The handheld device displays G6PD activity (in U/gHb) and hemoglobin (Hb) levels (in g/dL) two minutes after applying the blood buffer solution, however the manufacturer indicates that results cannot be considered if Hb readings are equal to or below 7g/dL. Field evaluation studies showed the Biosensor to have an accuracy of approximately 90% in identifying intermediate and deficient individuals when compared to spectrophotometry [21–23]. Since ease of use, time to diagnosis, and logistics and operational feasibility are preferable to spectrophotometry, the Biosensor has the potential to provide a quantitative G6PD measurement at the bed side and support point-of-

care diagnosis and treatment decisions [17]. The aim of this study was not to assess accuracy, but to determine the Biosensor's repeatability (assay precision when repeated under constant conditions) and reproducibility (assay precision under different conditions, such as across devices, operators and sites), since robust performance of these characteristics is necessary for rolling-out universal Biosensor thresholds for clinical decisions [24].

## Methods

### Ethics statement

IRB approvals or waivers were obtained from each participating institution prior to conducting the laboratory study. Risks to the technicians of using reconstituted human blood controls were discussed during the training and minimized by involving only technicians experienced in Good Laboratory Practice. Since no participants were enrolled, no informed consent was collected (S1 Table).

### Overview

Despite the Biosensor and reference method spectrophotometry being developed for fresh blood we had to use lyophilized and reconstituted standardized controls instead in order to ensure identical samples were used throughout the study period [21]. The study comprised of two phases (Fig 1). In the first phase (Phase A) baseline repeatability and inter-device reproducibility were defined under identical conditions. Ten Biosensors were tested repeatedly in parallel in a single laboratory by a single technician using commercial controls with a range of G6PD enzyme activity levels classified by the manufacturer as "High", "Intermediate", and "Low". Each control was tested 20 times over the course of five days with each Biosensor device and was tested in parallel by spectrophotometry and Hemocue. The study would only proceed to the second phase if mean repeatability of all Biosensors met or exceeded minimal requirements (see "statistical analysis" below). In the second phase (Phase B) reproducibility was assessed by shipping each device to a different, well-established laboratory. At each site, an identical set of controls was tested 40 times over the course of 10 days (120 measurements in total / site) by Biosensor and each reference assay, spectrophotometry and Hemocue. Reference methods were standardized as detailed below, and standard operating procedures were followed across all sites.

**Control samples.** To ensure that identical samples were tested across all sites, commercial controls were used, with all controls within one phase being from the same lot (Analytical Control Systems, Inc., Indiana, USA; S2 Table). ACS controls are routinely used to monitor quality of reference G6PD testing by spectrophotometry. They are derived from whole blood obtained from human donors in FDA licensed centers and pooled to represent high, intermediate, or low G6PD activity (Cat. Nos.: HC-108, HC-108IN, and HC-108DE respectively). ACS provide lot specific G6PD activity range guides and Hb estimates, these are based on automated spectrophotometry estimates conducted by Pointe-Scientific on the reconstituted controls (32 vials in total). ACS recommend that laboratories develop their own in-house ranges but provides guideline ranges per control category as well. We were unable to establish spectrophotometry-based ranges applicable to all sites due to the inherent site-specific variability of spectrophotometry (15), we therefore considered the manufacturer recommended ranges. ACS controls were provided in lyophilized form and reconstituted in each laboratory at standardized intervals. All reconstituted controls were stored at 4˚C and used within two days.

**Biosensor.** The SD Biosensor STANDARD G6PD test was performed following the manufacturer instructions. The assay uses single-use test strips (Cat. No. 02G6S10) that are inserted into the Biosensor (Cat. No. 02GA10). All test strips used in this study were from the same

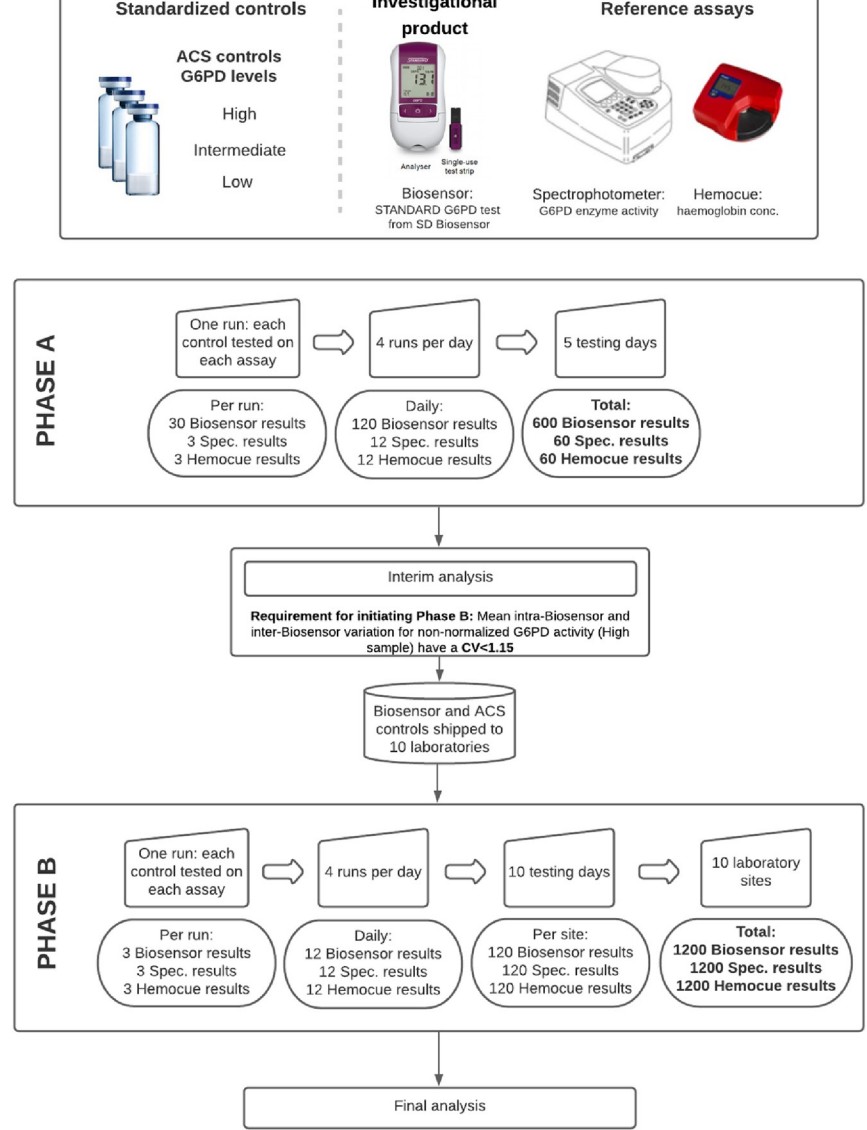

**Fig 1. Schematic overview of the study design.** CV: coefficient of variation. (Images reproduced with permission from SD Biosensor and Hemocue).

manufacturing lot (S2 Table). In brief, 10μl of the reconstituted control sample were mixed with the assay extraction buffer, then 10μl of the control-buffer solution were added to a single use test device that had already been inserted into the Biosensor. Sample transfer devices supplied with the Biosensor test, known as "Ezi tubes", were used at each step. The displayed normalized G6PD activity and Hb readings were recorded once the measurement was completed after the 2 mins running period.

Each Biosensor's functionality was checked daily with a multi-use STANDARD G6PD check strip, and sample testing only commenced if the quality control check was completed successfully. Every five days each Biosensor was also quality controlled with reconstituted control samples provided by SD Biosensor ("level 1" and "level 2"; Cat. No. 02G6C10). If results were outside of the recommended ranges for G6PD activity or Hb reading, the quality control

testing was repeated. If three consecutive quality control readings were outside the recommended ranges, testing with the respective Biosensor device was aborted.

**Spectrophotometry and Hemocue.**   Spectrophotometry was performed using kits from Pointe Scientific (Michigan, USA; Cat. No. G7583) according to the manufacturer's recommendations. The brand of spectrophotometer varied between laboratories, but all instruments were temperature-controlled, cuvette based, and measured absorption at 340 nm. Sample absorbance was measured at 0 and 5 minutes at 37°C and the difference in absorbance was used to calculate G6PD activity in U/dL following a standard formula provided by the Pointe-Scientific assay manufacturer. Measurements were run in duplicate (i.e. the sample reaction was divided into 2 separate cuvettes and measured independently) and the mean of the two G6PD results was recorded. If the coefficient of variation of the two measures exceeded 15% (CV>0.15), a third measurement was required. G6PD activity was then normalized by a Hb reading (Hemocue 201, 301 or 801, Angelholm, Sweden) to generate G6PD activity in U/gHb. Hemocue devices were used following manufacturer instructions, and the Hb reading was recorded separately.

**Training.**   All technicians conducting the experiments had at least a bachelor's degree or higher, several years' experience of working in a laboratory and familiarity with spectrophotometry, however not necessarily with the G6PD assay used in this study. Each technician received standardized training in an online session and had to pass a Biosensor proficiency test prior to conducting the experiments. In each laboratory, a single technician performed the analyses with each diagnostic across all study testing days.

**Statistical analysis.**   Data were recorded on standardized forms and then transferred to an Excel database (Microsoft Corp, Washington, USA), with standard data entry cross-checks. Analysis was undertaken using Stata version 15 (Stata Corp, College Station, Texas, USA).

Depending on data distribution, summary findings were displayed as mean or median with 95% confidence intervals or interquartile range (IQR) respectively. Repeatability and reproducibility were assessed by linear, random effects, and mixed effects regression models as appropriate, and by calculating coefficients of variation (CV). Spearman's Rank coefficient ($r_s$) was calculated to determine the correlation between Biosensor and reference method. Absolute differences between experimental (Biosensor) and reference assays (spectrophotometer and Hemocue) were assessed by Bland Altman plots and the Wilcoxon matched pairs signed rank test. The study only progressed to Phase B if the mean CV for the High control sample across all devices was less than 0.150 [25]. Combined mean difference and correlation coefficients were calculated for Phase A where one spectrophotometry reading served as reference for all ten devices. In Phase B findings were not combined since each site performed their own reference measurement and the reference method for G6PD activity is known to show significant variation, not allowing for a direct comparison [15].

Following Bonferroni correction, the level of significance was set at p<0.005 whenever multiple comparisons were done.

## Results

### Phase A

Results from the ten Biosensor devices tested with High and Intermediate controls were available for five consecutive days, and Low controls for four days due to limited stocks of same-lot Low controls.

Hb-normalized G6PD activities did not differ significantly between Biosensor devices (Fig 2 and S3 Table, p = 1.000). However, Hb readings differed significantly (p<0.001, adjusted $R^2$ = 0.121). Compared to device 1 (baseline), devices 3 and 9 had significantly lower Hb

readings with a difference of 0.477 g/dL (p = 0.007) and 0.623 g/dL (p<0.001) respectively, while device 5 had a significantly higher Hb result with a difference of 0.665 g/dL (p<0.001) when comparing readings from all three controls (Fig 3 and S3 Table).

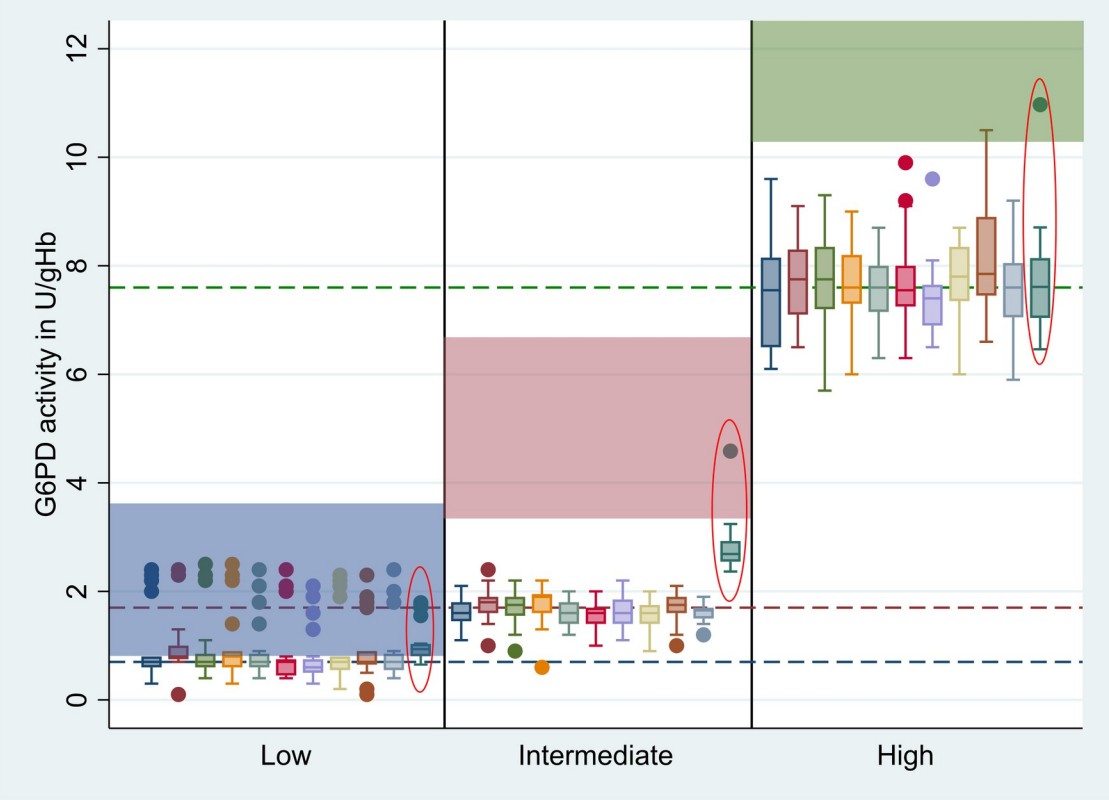

**Fig 2. G6PD activity measured by each Biosensor device and spectrophotometry.** Red circled bars = reference method (Hb normalized spectrophotometry result), Blue shade = recommended range for ACS Low controls, Red shade = recommended range for ACS Intermediate controls, Green shade = recommended range for ACS High controls, Blue dotted line = median activity of ACS Low controls across all devices excluding spectrophotometry, Red dotted line = median activity of ACS Intermediate controls across all devices excluding spectrophotometry, Green dotted line = median activity of ACS High controls across all devices excluding spectrophotometry, dots represent outliers.

Each run included a spectrophotometer measurement which was matched with a result from each of the 10 Biosensor devices (Fig 1). The Hb-normalized G6PD activity readings of the Biosensor and spectrophotometry were positively correlated across all three control categories ($r_s$ = 0.859, p<0.001), however median readings differed significantly for Low (mean difference: -0.1 U/gHb, 95% limit of agreement [95%LoA]: -0.8 to 0.5, p<0.001) and Intermediate controls (mean difference: -1.2 U/gHb, 95%LoA: -2.3 to -0.1, p<0.001) while median activities did not differ significantly for High controls (mean difference: -0.1 U/gHb, 95%LoA: -2.3 to 2.1, p = 0.554). Hb readings from the Biosensor and Hemocue showed a significant correlation in five out of 10 devices (p<0.005; Table 1).

Since normalized G6PD activity did not differ significantly between Biosensor devices, results were pooled and compared to spectrophotometry by Bland-Altman plot. The G6PD readings of the Biosensor were significantly lower (mean difference: 0.5U/gHb, 95%LoA: -2.2 to 1.3; p<0.001) compared to spectrophotometry (Fig 4). Mean Hb readings using the Biosensor were 1.8g/dL (95%LoA: -4.7 to 1.1) lower than those of the Hemocue (p<0.001) (S1 Fig).

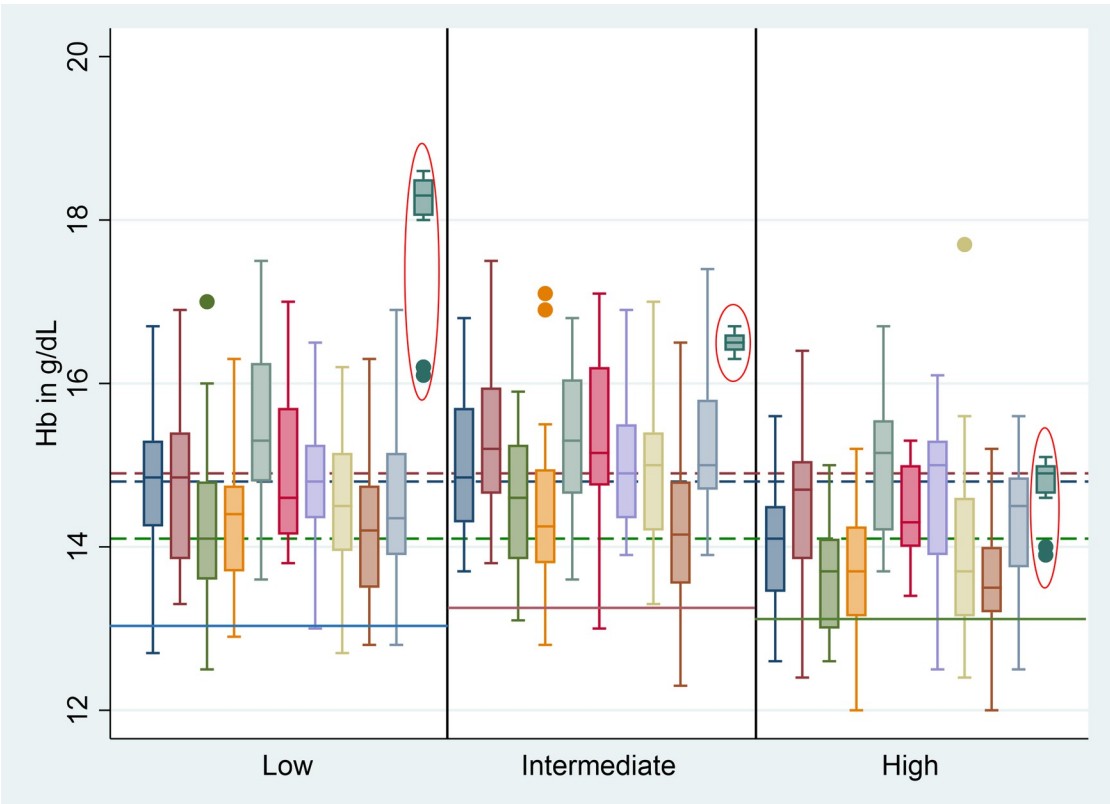

**Fig 3. Hb levels measured by each Biosensor device and Hemocue.** Red circled bars = reference method (Hemocue), Blue line = recommended point estimate for ACS Low controls, Red line = recommended point estimate for ACS Intermediate controls, Green line = recommended point estimate for ACS High controls, Blue dotted line = median Hb reading for Low controls, Red dotted line = median Hb reading for Intermediate controls, Green dotted line = median Hb reading for High controls, dots represent outliers.

**Table 1. Correlation of Biosensor and reference method, Phase A.**

| Device | G6PD ($r_s$, $p^*$) | Hb ($r_s$, $p^*$) |
|---|---|---|
| 1 | 0.871, <0.001 | 0.427, <0.001 |
| 2 | 0.884, <0.001 | **0.278, 0.031** |
| 3 | 0.861, <0.001 | 0.424, <0.001 |
| 4 | 0.839, <0.001 | **0.288, 0.025** |
| 5 | 0.874, <0.001 | 0.416, 0.001 |
| 6 | 0.868, <0.001 | **0.314, 0.015** |
| 7 | 0.874, <0.001 | **0.183, 0.163** |
| 8 | 0.856, <0.001 | 0.394, 0.002 |
| 9 | 0.885, <0.001 | 0.384, 0.003 |
| 10 | 0.871, <0.001 | **0.322, 0.012** |
| Pooled | 0.868, <0.001 | 0.318, <0.001 |

Highlighted correlations are not significant

$^*$level of significance set at p<0.005 following Bonferroni correction

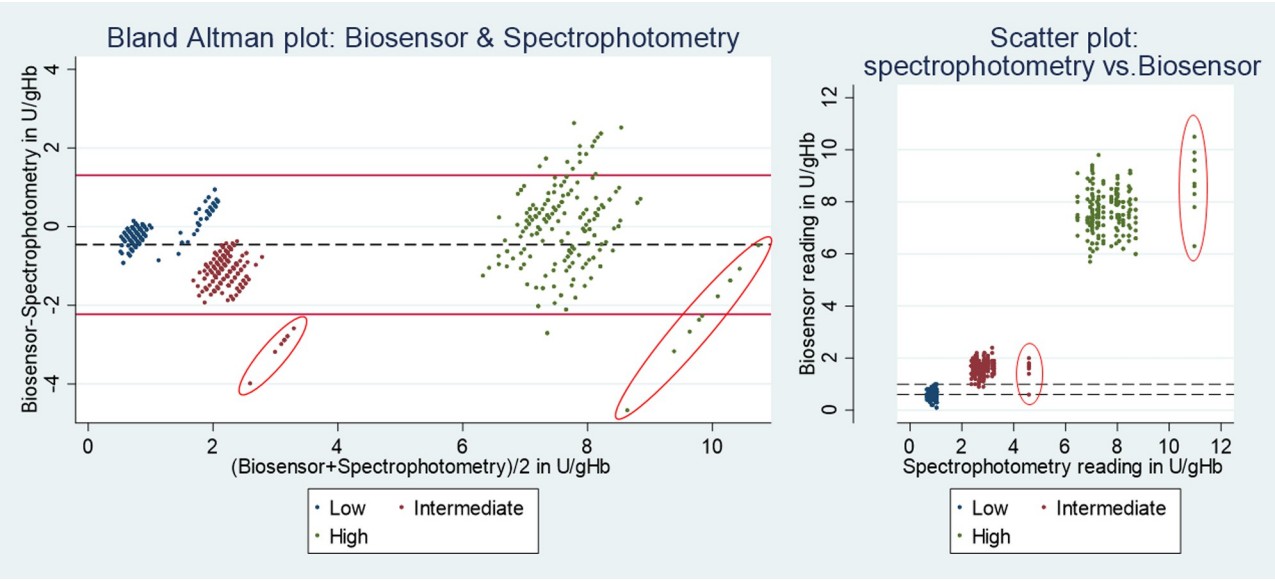

**Fig 4.** Phase A: G6PD activity measured by Biosensor and spectrophotometry. Red circled results are influential points generated by spectrophotometry; Left: Red horizontal lines indicate upper and lower end of 95% limit of agreement, black dotted line indicates mean difference. Right: horizontal dotted lines indicate the overlap between Low and Intermediate readings by Biosensor.

Median G6PD readings from the Low control were below the manufacturer recommended range (0.8U/gHb– 3.8U/gHb) when measured by Biosensor (0.7U/gHb, interquartile range (IQR): 0.6–0.8) but not by spectrophotometry (0.9U/gHb, IQR: 0.8–1.0), while median G6PD readings for Intermediate (Biosensor: 1.7U/gHb, IQR: 1.5–1.8; spectrophotometry: 2.7, IQR: 2.5–2.9) and High controls (Biosensor: 7.6U/gHb, IQR: 7.5–8.2; spectrophotometry: 7.6U/gHb, IQR: 7.0–8.1) were below the recommended ranges (Intermediate: 3.7U/gHb– 6.8U/gHb; High 10.3U/gHb– 19.1U/gHb respectively) for both assays. Neither the median Hb readings of the Biosensor nor Hemocue were equal to the point estimate provided by the control sample manufacturer (Tables 2 and S3 and Figs 2 and 3).

Median readings of High, Intermediate, and Low controls were distinct by spectrophotometry (High vs. Intermediate: p<0.001 and Intermediate vs. Low: p<0.001), but while median readings by Biosensor also differed significantly between all three control categories (all p<0.001) six Intermediate readings overlapped with 122 Low results. All six Intermediate readings were generated by different devices and during different testing runs. Influential outliers generated by two spectrophotometry readings were identified visually (Fig 4).

The median CV across all Biosensor G6PD measurements for High controls was 0.111, below the pre-defined acceptability threshold of 0.150, while the CV for Hb measurement was below 0.070 for all controls (Table 2). The study therefore proceeded to Phase B.

**Table 2. Phase A results: coefficient of variation and median value measured for all Biosensors combined, spectrophotometry and Hemocue.** Summaries are for all Phase A results combined.

| Assay | G6PD activity (in U/gHb) | | | Hb (in g/dL) | | |
|---|---|---|---|---|---|---|
| | Low | Intermediate | High | Low | Intermediate | High |
| Biosensors (n = 600) | | | | | | |
| CV | 0.260 | 0.172 | 0.111 | 0.061 | 0.068 | 0.069 |
| Median | 0.7 | 1.7 | 7.6 | 14.8 | 14.9 | 14.1 |
| IQR | 0.6–0.8 | 1.5–1.8 | 7.2–8.2 | 14.3–15.6 | 14.2–15.5 | 13.5–15.0 |

*(Continued)*

**Table 2.** (Continued)

| | | | | | | |
|---|---|---|---|---|---|---|
| SD | 0.17 | 0.28 | 0.85 | 0.91 | 1.01 | 0.98 |
| Range | 0.1–1.0 | 0.6–2. 4 | 5.7–10.5 | 12.9–17.5 | 12.3–17.5 | 12.0–17.7 |
| **Reference (n = 60)** | | | | | | |
| CV | 0.141 | 0.165 | 0.124 | 0.009 | 0.007 | 0.022 |
| Median | 0.9 | 2.7 | 7.6 | 18.4 | 16.5 | 14.9 |
| IQR | 0.8–1.0 | 2.5–2.9 | 7.0–8.1 | 18.2–18.5 | 16.4–16.6 | 14.7–15.0 |
| SD | 0.12 | 0.47 | 0.96 | 0.01 | 0.01 | 0.02 |
| Range | 0.7–1.0 | 2.4–4.6 | 6.4–11.0 | 18.0–18.6 | 16.3–16.7 | 13.9–15.1 |
| | **Recommended range G6PD (U/gHb)** | | | **Recommended value Hb (g/dL)** | | |
| **Manufacturer recommendation** | 0.8–3.8 | 3.7–6.8 | 10.3–19.1 | 13.0 | 13.2 | 13.1 |

CV = coefficient of variation, IQR = interquartile range, SD = standard deviation

## Phase B

Biosensors and controls were shipped to one laboratory in Bangladesh, Brazil, France and Indonesia respectively, two laboratories in Thailand, and four in the USA for further testing. Staff at six laboratories had previous experience with the Biosensor. Pooled readings from Low controls were within the recommended range (0.8U/gHb– 3.8U/gHb) by Biosensor (median: 2.2U/gHb, IQR: 2.0–2.4) and spectrophotometry (1.9U/gHb, IQR: 1.6–2.2). However Intermediate pooled readings by Biosensor (1.9U/gHb, IQR: 1.6–2.1) and spectrophotometry (3.1U/gHb, IQR: 2.7–3.5) were below the manufacturer recommended range (3.7U/gHb– 6.8U/gHB), as were the readings of High controls (Biosensor: 8.0U/gHb, IQR: 7.4–8.7; spectrophotometry: 8.2U/gHb, IQR: 7.4–9.2; recommended range 10.3U/gHb– 19.1 U/gHb). Observed ranges across all categories were greater for spectrophotometry than for the Biosensor (S4 Table).

The correlation between Biosensor and reference method was significant and positive for G6PD readings (in U/gHb), while Hb readings of five Biosensor devices did not correlate significantly with the Hemocue at the 0.5% (p<0.005) significance level (Table 3).

Mean G6PD readings by Biosensor and spectrophotometry differed significantly in five of 10 sites, while Hb readings showed a significant difference between Biosensor and Hemocue

**Table 3. Mean difference and correlation of Biosensor and reference method by site in phase B.**

| Site | G6PD (U/gHb): Mean Difference (95% LoA, p) | G6PD ($r_s$, p) | Hb (g/dL): Mean Difference (95% LoA, p) | Hb ($r_s$, p) |
|---|---|---|---|---|
| 1 | -0.1 (-1.8 to 1.7, 0.245) | 0.548, <0.001 | **-1.9 (-4.0 to 0.1, <0.001)** | **0.254, 0.005**[*] |
| 2 | -0.2 (-1.8 to 1.4, 0.004) | 0.564, <0.001 | **-1.9 (-4.1 to 0.2, <0.001)** | **-0.023, 0.806**[*] |
| 3 | 0.2 (-1.7 to 2.0, 0.088) | 0.659, <0.001 | **-3.0 (-4.9 to -1.2, <0.001)** | 0.350, <0.001 |
| 4 | 0.1 (-2.0 to 2.2, 0.504) | 0.536, <0.001 | **-2.5 (-4.7 to -0.3, <0.001)** | 0.312, 0.001 |
| 5 | **1.1 (-1.9 to 4.2, <0.001)** | 0.575, <0.001 | **0.6 (-0.8 to 2.0, <0.001)** | 0.416, <0.001 |
| 6 | -0.3 (-3.2 to 2.7, 0.055) | 0.509, <0.001 | **-2.6 (-5.9 to 0.6, <0.001)** | **0.095, 0.303**[*] |
| 7 | **-1.1 (-3.1 to 0.8, <0.001)** | 0.514, <0.001 | -0.1 (-1.7 to 1.5, 0.133) | **0.239, 0.009**[*] |
| 8 | **-0.9 (-2.8 to 1.0, <0.001)** | 0.609, <0.001 | **-1.5 (-3.4 to 0.4, <0.001)** | 0.500, <0.001 |
| 9 | **-2.6 (-7.6 to 2.3, <0.001)** | 0.569, <0.001 | -0.6 (-3.0 to 4.3, 0.009) | 0.391, <0.001 |
| 10 | **-1.0 (-3.1 to 1.1, <0.001)** | 0.563, <0.001 | **-0.9 (-3.0 to 1.1, <0.001)** | **0.211, 0.021**[**] |

95% LoA = 95% limit of agreement

[*] Correlation was non-significant across all control categories (all p>0.005)

[**]Only correlation for Intermediate controls was significant: $r_s$ = 0.580, p = 0.001

in eight of 10 sites. Observed mean differences between Biosensor and spectrophotometry ranged from -2.6U/gHb to +1.1U/gHb across the ten devices and from -3.0 g/dL to 0.6g/dL between Biosensor and Hemocue (Table 3).

Low and Intermediate controls could not be differentiated by Biosensor, in fact median readings of Intermediate controls (1.9U/gHb, IQR: 1.6–2.1) were significantly lower than median Low readings (2.2U/gHb, IQR: 2.0 to 2.4, p<0.001), a trend that was seen across all sites. In contrast High controls (8.0U/gHb, IQR 7.4–8.7) were clearly distinct from the other controls across all sites (p<0.001) (Figs 5 and S2–S5, and S4 Table).

Median spectrophotometry measures for Low (1.9U/gHb, IQR: 1.6–2.2) and Intermediate (3.1U/gHb, IQR: 2.7–3.5) readings differed significantly (p<0.001), as did Intermediate and High (8.3U/gHb, IQR: 7.4–9.3) controls (p<0.001). This trend and level of significance was consistent within all sites but not between sites (Figs 5 and S3–S6, and S4 Table).

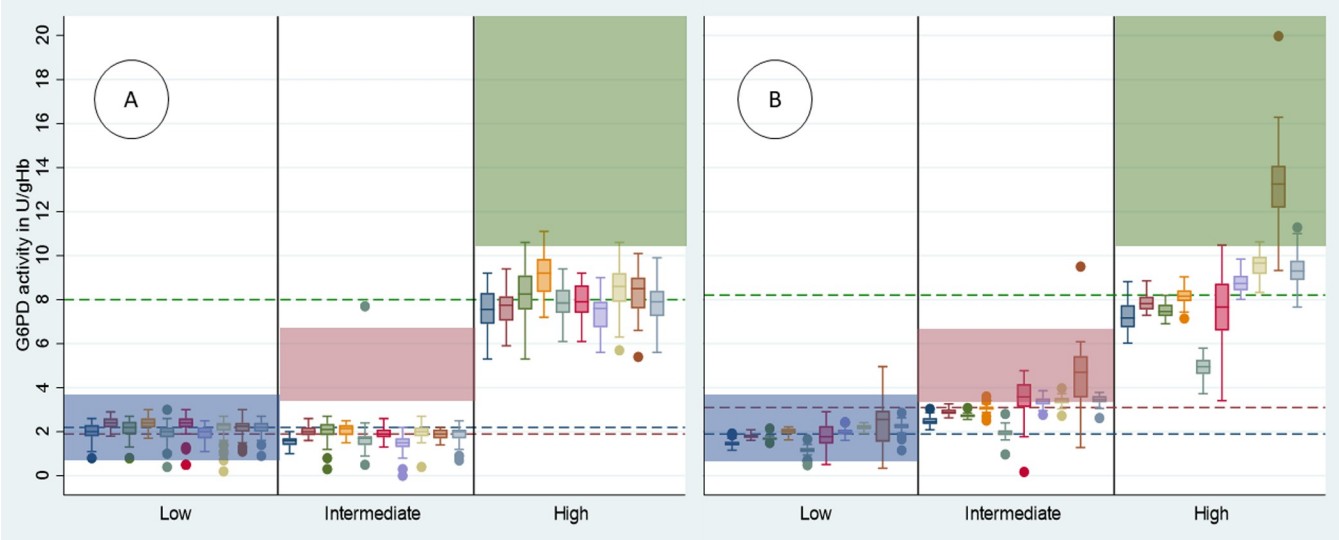

**Fig 5.** Box and whisker plot of G6PD activity / control by Biosensor (A) and spectrophotometry (B) across Phase B sites. Blue shade = recommended range for ACS Low controls, Red shade = recommended range for ACS Intermediate controls, Green shade = recommended range for ACS High controls, Blue dotted line = median activity of ACS Low controls across all devices, Red dotted line = median activity of ACS Intermediate controls across all devices, Green dotted line = median activity of ACS High controls across all devices, dots represent outliers.

Repeatability (within-site assay precision, as measured by CV) varied more between sites for spectrophotometry than the Biosensor (Fig 5). Site-level CVs of the Biosensor for the High control ranged from 0.103 to 0.125 (SD: 0.009), while spectrophotometry results ranged from 0.050 to 0.137 (SD: 0.043) (S5 Table).

Normalized G6PD activities did not differ significantly between Biosensors (p = 0.436), however spectrophotometry readings showed significant variation by site (p<0.001) (Figs 5, S2, S3, S4, S5 and S6). Hb readings differed significantly by Biosensor across all sites (p<0.001) and variation was even greater for the Hemocue (p<0.001) (S5 Table).

## Comparing Phases A and B

The CVs between Phase A and Phase B did not differ significantly (p = 0.201). The Intermediate and High controls in Phase A and B were from the same manufacturing lots so Biosensor readings could be compared directly (S2 Table). G6PD activities differed significantly between

**Table 4. Mean difference in G6PD activity for Intermediate and High controls by each Biosensor device between Phases A and B.** Low controls were not directly comparable between Phases as these were from different lots.

| Device | Intermediate: mean difference in U/gHb*, (95% CI, p) | High: mean difference in U/gHb*, (95% CI, p) |
|:---:|:---:|:---:|
| **1** | -0.1 (-0.1 to 0.2, 0.481) | 0.0 (-0.5 to 0.5, 0.993) |
| **2** | -0.3 (-0.4 to -0.1, <**0.001**) | 0.1 (-0.3 to 0.5, 0.657) |
| **3** | -0.3 (-0.5 to -0.1, **0.016**) | -0.5 (-1.1 to 0.0, 0.050) |
| **4** | -0.3 (-0.5 to -0.2, <**0.001**) | -1.5 (-2.1 to -1.0, <**0.001**) |
| **5** | -0.1 (-0.6 to 0.3, 0.559) | -0.2 (-0.6 to 0.2, 0.399) |
| **6** | -0.3 (-0.4 to -0.2, <**0.001**) | -0.2 (-0.7 to 0.2, 0.326) |
| **7** | 0.1 (-0.1 to 0.3, 0.184) | 0.0 (-0.4 to 0.4, 0.906) |
| **8** | -0.4 (-0.6 to -0.2, <**0.001**) | -0.8 (-1.3 to -0.3, **0.003**) |
| **9** | -0.2 (-0.3 to 0.0, **0.032**) | -0.1 (-0.7 to 0.5, 0.765) |
| **10** | -0.2 (-0.4 to -0.04, **0.018**) | -0.3 (-0.8 to 0.2, 0.224) |
| **Pooled** | -0.2 (-0.3 to -0.1, <**0.001**) | -0.4 (-0.5 to -0.2, <**0.001**) |

*Phase A–Phase B

Phase A and B (p<0.001) and while Intermediate control readings in Phase A were significantly higher for eight of the 10 devices, the difference did not exceed 0.4U/gHb. For High controls, a significant difference was observed in three devices with a maximum difference of 1.5U/gHb (Table 4).

## Discussion

The reproducibility (inter-device precision) of the Biosensor did not differ significantly between devices, either when handled by the same technician (Phase A), when operated in different settings by different end users (Phase B), or when the same device was handled by different operators (Phase A vs B). In contrast there was significant variation when G6PD activity was measured by spectrophotometry between sites despite standardized controls and procedures, a phenomenon that has been reported previously [15].

Four out of the ten participating sites had not used the Biosensor previously. Following standardized online training, all sites were able to generate G6PD measurements with good precision that did not differ significantly between sites. Precision of the spectrophotometry results was more variable between sites, with some sites exceeding Biosensor repeatability while others had lower precision. There was a good correlation between the Biosensor and spectrophotometry results when these were assessed in a single lab in Phase A. However, while spectrophotometry could discriminate reliably between the three control types, the Biosensor results less clearly distinguished Low from Intermediate controls, with six of the 200 repeat measurements overlapping. Correlation between Biosensor and spectrophotometry was lower in Phase B when devices were assessed in different laboratories due to the variability of the spectrophotometry. In Phase B the Biosensor did not distinguish between Low and Intermediate controls. In fact, the results were significantly lower for Intermediate compared to Low controls and this was consistent across all sites; in contrast spectrophotometry in Phase B was able to distinguish between all three control categories.

Besides G6PD activity, the Biosensor also measures and displays Hb concentration. The repeatability of Hb measurements by Biosensor was better than by Hemocue, with best inter-device repeatability observed when either device was operated by a single user. Hb readings of both devices correlated poorly, not least since the recommended Hb point estimates for all three controls were very similar. Absolute pooled readings for the Biosensor were 1.8g/dl

lower in Phase A compared to paired readings of the Hemocue, however readings from the Biosensor were closer to the recommended point estimate suggested by the ACS manufacturer. Determining accuracy of Biosensor Hb readings against the Hemocue reference assay with reconstituted lyophilised controls is of limited clinical relevance. A study from the US compared paired Hb measurements from fresh, venous, samples by Biosensor and Hemocue (model 201+) and found the mean difference to be 1.0g/dL [22]; a study comparing Biosensor Hb readings from venous blood samples to the results of a complete blood count (CBC), found readings to differ by 0.4g/dL [21], and a recent study from Brazil found the mean difference again to be less than 1 g/dl [23].

Our findings have several limitations. G6PD activity and Hb levels were measured in commercial lyophilised controls to ensure cross-laboratory standardisation, but the Biosensor and reference assays are developed for testing fresh venous or capillary blood. Stabilizing agents contained in the controls may have affected the Biosensor and/or reference method and this effect may differ by assay. Repeatability and reproducibility of each assay should not have been impacted by this difference, but it may have affected accuracy of either device which is best assessed using fresh blood samples [21, 22]. While providing reference ranges, the manufacturer ACS suggests developing in house reference ranges for all controls. We were unable to establish spectrophotometry-based ranges applicable to all sites due to the inherent site-specific variability of spectrophotometry [15] and instead considered the ranges provided. This approach likely explains why G6PD readings generated by Biosensor and spectrophotometry were below the ACS manufacturer's recommended range and Hb readings by either assay did not match the guideline point estimates. This was consistent across different devices when assessed by a single user and by different laboratories. Unfortunately, the supplier was unable to provide additional controls from the same lots for further testing to clarify this issue. The observed narrow activity ranges meant that there was little difference in activity levels between the Intermediate and Low controls, limiting the activity range that was assessed. Two spectrophotometry readings for Intermediate and High controls appeared to be outliers in Phase A, both of which were included since readings were within the recommended range and this may also have reduced the correlation between assays and the derived absolute difference. Although sites used different Hemocue models (Hemocue 201, 301 and 801), results from all devices were pooled which may have resulted in an increase in variability for the Hb reference. Finally, all measurements were done by highly qualified technicians in a research setting, not reflective of a real-world scenario, accordingly reproducibility of the Biosensor may be lower when used in a clinical setting.

The precision of the Biosensor demonstrated in this study, and the good accuracy reported from field and other evaluation studies [21–23, 26], indicate that the Biosensor could be a valuable quantitative point-of-care diagnostic; however, we found that spectrophotometry, when performed well, remains the gold standard with precision superior to the Biosensor. The reproducibility observed in this study indicates that the technology is likely to permit direct comparison of results generated by different Biosensor devices and trained users [15]. If confirmed in clinical settings, the Biosensor has the potential to be an important tool to facilitate the broader roll out of 8-aminoquinoline radical cure. Clinical data will be important to further investigate the poor discriminatory power of the Biosensor at low and intermediate G6PD activities observed with the lyophilised samples, however given that the Biosensors' most probable designation will be to distinguish G6PD normal individuals from those with less than normal activity (at a cut-off of 70% activity for Tafenoquine) the observed poor discriminatory power at lower activities is unlikely to be of significant practical relevance. Finally, it will be important to verify whether the observed precision demonstrated here is maintained when the device is operated under routine conditions and in anaemic patients, as well as to define

training requirements for intended users at the point-of-care. In conclusion, our findings suggest that the Biosensor offers reproducible quantitative diagnosis of G6PD status at the point-of-care in the hands of well-trained technicians. If repeatability and reproducibility as well as the previously reported accuracy are confirmed under real life conditions, the Biosensor has the potential to simplify access to effective radical cure of *P. vivax* malaria.

## Supporting information

**S1 Data. Underlying database.**
(XLSX)

**S1 Fig. Bland Altman Plot: Hb measured by Biosensor and Hemocue.** *Red horizontal lines indicate upper and lower end of 95% limit of agreement, green line indicates mean difference.*
(TIF)

**S2 Fig. G6PD activity/ site / control by Biosensor.**
(TIF)

**S3 Fig. Low Controls: G6PD activity / site / assay.**
(TIF)

**S4 Fig. Intermediate Controls: G6PD activity / site / assay.**
(TIF)

**S5 Fig. High Controls: G6PD activity / site / assay.**
(TIF)

**S6 Fig. G6PD activity / site / control by Spectrophotometry.**
(TIF)

**S1 Table. Ethics boards of participating sites.**
(DOCX)

**S2 Table. Lot numbers of ACS controls per phase.**
(DOCX)

**S3 Table. Coefficient of variation and median G6PD activity / Hb concentration measured per assay and device in Phase A.**
(DOCX)

**S4 Table. Summary of readings for Biosensor (G6PD activity and Hb), spectrophotometry, and Hemocue across 10 sites in Phase B.**
(DOCX)

**S5 Table. Phase B coefficient of variation (CV) of G6PD measurements by Biosensor vs Spectrophotometry and Haemoglobin measurements by Biosensor vs Hemocue.**
(DOCX)

## Acknowledgments

All STANDARD G6PD analysers and corresponding consumables (test strips) were provided by the manufacturer SD Biosensor at no cost. The authors thank the logistics and procurement teams across their institutions for support with implementing the study shipments.

The presented material has been reviewed by the Walter Reed Army Institute of Research. There is no objection to its presentation and/or publication. The opinions or assertions

contained herein are the private views of the authors and are not to be construed as official or as reflecting true views of the Department of the Army or the Department of Defense. The investigators have adhered to the policies for protection of human subjects as prescribed in AR 70 to 25.

## Author Contributions

**Conceptualization:** Benedikt Ley, Mohammad Shafiul Alam, Rosalind E. Howes.

**Data curation:** Benedikt Ley.

**Formal analysis:** Benedikt Ley, Stefano Ongarello.

**Funding acquisition:** Rosalind E. Howes.

**Investigation:** Ari Winasti Satyagraha, Mohammad Golam Kibria, Jillian Armstrong, Germana Bancone, Amy K. Bei, Greg Bizilj, Marcelo Brito, Michael E. von Fricken, Gornpan Gornsawun, Brandon Lam, Didier Menard, Wuelton Monteiro, Sampa Pal, Lydia Visita Panggalo, Sunil Parikh, Alessandra da Silva Orfano, Martina Wade, Mariusz Wojnarski, Kuntawunginn Worachet, Aqsa Yar, Mohammad Shafiul Alam.

**Methodology:** Benedikt Ley, Ari Winasti Satyagraha, Mohammad Golam Kibria, Xavier C. Ding, Gonzalo J. Domingo, Mohammad Shafiul Alam, Rosalind E. Howes.

**Project administration:** Benedikt Ley, Rosalind E. Howes.

**Supervision:** Benedikt Ley, Ari Winasti Satyagraha, Michael E. von Fricken, Rosalind E. Howes.

**Writing – original draft:** Benedikt Ley.

**Writing – review & editing:** Benedikt Ley, Ari Winasti Satyagraha, Mohammad Golam Kibria, Jillian Armstrong, Germana Bancone, Amy K. Bei, Greg Bizilj, Marcelo Brito, Xavier C. Ding, Gonzalo J. Domingo, Michael E. von Fricken, Brandon Lam, Didier Menard, Wuelton Monteiro, Stefano Ongarello, Sampa Pal, Sunil Parikh, Daniel A. Pfeffer, Ric N. Price, Alessandra da Silva Orfano, Martina Wade, Mariusz Wojnarski, Mohammad Shafiul Alam, Rosalind E. Howes.

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
