## [Decision Letter · Decision Letter 0]

19 Aug 2021

Dear Dr. Ley,

Thank you very much for submitting your manuscript "The STANDARD G6PD test (SD Biosensor) shows good repeatability and reproducibility in a multi-laboratory comparison" for consideration at PLOS Neglected Tropical Diseases. As with all papers reviewed by the journal, your manuscript was reviewed by members of the editorial board and by several independent reviewers. In light of the reviews (below this email), we would like to invite the resubmission of a significantly-revised version that takes into account the reviewers' comments. 

We cannot make any decision about publication until we have seen the revised manuscript and your response to the reviewers' comments. Your revised manuscript is also likely to be sent to reviewers for further evaluation.

Sincerely,

J. Kevin Baird

Guest Editor

Mary Lopez-Perez

Deputy Editor

Reviewer's Responses to Questions

**Key Review Criteria Required for Acceptance?**

**Methods**

-Are the objectives of the study clearly articulated with a clear testable hypothesis stated?

-Is the study design appropriate to address the stated objectives?

-Is the population clearly described and appropriate for the hypothesis being tested?

-Is the sample size sufficient to ensure adequate power to address the hypothesis being tested?

-Were correct statistical analysis used to support conclusions?

-Are there concerns about ethical or regulatory requirements being met?

Reviewer #1: This manuscript reports on and intra-laboratory and and inter-laboratory analysis of G6PD assays carried out by a commercial device called BIOSENSOR. 

The results show that, overall, the agreement among multiple BIOSENSOR devices, is good, but with exceptions. The comparison among several distant laboratory must have posed considerable challenges and it is an enterprise for which the Authors are to be commended. 

However, I have to raise several general criticisms.

1. The main motive for this study, as stated in the introduction and elsewhere, is to avoid hemolytic complications from primaquine or tafenoquine. The assessment of G6PDd in males is very easy: it does not require either a spectrophotometric assay or BIOSENSOR, as it can be done by the fluorescent spot test or by other inexpensive screening tests. The need for a quantitative tests regards only females. Surprisingly, this is not stated anywhere. 

2. In view of the above, in order to assess BiOSENSOR the focus should be on the ‘intermediate’ group: specifically, on the success rate of the device in detecting heterozygotes with a range of G6PD activities. Unfortunately, by using a single intermediate control the work was not designed to assess this success rate. In addition, it is precisely in the intermediate group that problems were encountered.

3. The Authors should briefly explain how the BIOSENSOR works. Is G6PD measured by NADPH production, or by formazan production, or in what other way? Is Hb measured by cyanmethemoglobin production or in what other way?

4. Lyophilization is not a standard way to process blood samples for a G6PD assay (and it may be a factor in some of the problems encountered). Given the design of their study, the Authors should show what was the agreement between G6PD values obtained in one lab on a set of samples, fresh versus after lyophilisation and re-constitution; and on 0 to 48 hours after re-constitution. 

5. In heterozygous (intermediate) females what matters from the point of view of hemolysis is, rather than the level of G6PD activity in a hemolysate, the proportion of G6PDd red cells within their red cell mosaicism. Of course the two do correlate, but somewhere in the paper this important fact should be mentioned.

Reviewer #2: The overall study design is appropriate, allowing direct comparison between sites. However, as the authors also mentioned, the devices used in this study (Biosensor and Hemocue) all require the use of fresh blood. So the use of lyophilised RBCs may not faithfully reflect the real situations.

Reviewer #3: The objectives of the study are very clearly articulated, the design is appropriate, the sample set and size are clearly described and justified, the statistical analysis is robust and there are no ethical or regulatory concerns that I can tell.

There are no major additional analyses/experiments required.

**Results**

-Does the analysis presented match the analysis plan?

-Are the results clearly and completely presented?

-Are the figures (Tables, Images) of sufficient quality for clarity?

Reviewer #1: Line 145. It would be important to know exactly what these controls are: I could not find any G6PD controls on the ACS site. What was the original G6PD activity on the fresh blood sample of each? I think

Here and throughout the manuscript I assume ‘high’ means a sample from a G6PD normal person: why use the term high? 

Fig. 2. In this figure there are several strange features on which there is no comment in the text until the discussion, and the legend is incomplete: are all the cricles outliers? Shaded areas are said to be the recommended ranges (presumably recommended by ACS). I could not find what were the original G6PD activity values of the control samples provided; but even the spectrophotometric assay gives values way below recommended ranges in the intermediate and ‘high’ areas. This suggests that the lyophilization creates a problem. The red dotted line is said to be the median of the low controls, but I think it is the median of the intermediate controls. Similar apparent mistakes in defining the horizontal lines are replicated in the captions of Figs. 5 and 6. 

Fig. 3. In the Hb measurement the agreement among devices is clearly not very good: whatever the overall statistics say, most of the data from device number 5 (from the left) are above most of the data from device number 4. Since I expect Hb to be independent of G6PD, I don’t understand why the reference (Hemocue) is way higher than BIOSENSORs in low and intermediate, and about right in ‘high’.

Fig. 4. The overall correlation between BIOENSOR and spectrophotometry is obviously OK; but the fact remains that in the intermediate group the bulk is about 3 IU/G Hb by spectrophotometry, and it is instead about 2 by BIOSENSOR. The superiority of spectrophotometry is supported by Suppl figure S4.

Fig. 5. In phase B, by the Authors’ own admission, the low values of the ‘high’ controls and the lack of discrimination between low and intermediate controls are a bit of a disaster. 

Fig. 6. In spite of considerable scattering of values among different labs, the results are clearly better by spectrophotometry than by BIOSENSOR.

Reviewer #2: Yes.

Reviewer #3: The results are clearly and completely presented and the supplementary data adds to the manuscript.

The figures and tables are of sufficient quality and clarity, I would however suggest to combine figures 2 & 3 and figures 5 & 6 in a panel figures so as to make direct comparison easier for the reader. 

In addition, I would revise the tables and condense the information into fewer tables/add some of the tables to the supplementary data.

**Conclusions**

-Are the conclusions supported by the data presented?

-Are the limitations of analysis clearly described?

-Do the authors discuss how these data can be helpful to advance our understanding of the topic under study?

-Is public health relevance addressed?

Reviewer #1: Overall, the Authors have given a reasonably critical account of the shortcomings of BIOSENSOR emerging from their study. The emphasis on reproducibility, related to precision, cannot overcome the rather glaring problem with accuracy. 

The statement on lines 388-393 suggests that using lyophilized material was a weak point in the design of the study.

Line 414 onwards. It is a bit of an anti-climax that, having conducted a deliberate study on agreement among multiple sites, the authors have to quote previous studies on the reliability of BIOSENSOR; and they then proceed to state that their work needs to be “confirmed in clinical settings”, after it has been found wanting in experienced laboratory settings.

Reviewer #2: Yes.

Reviewer #3: The conclusions are supported by the data and the limitations of the analysis are described in some detail. The authors briefly touch on how the data can inform the topic under study and briefly touch on public health relevance.

**Editorial and Data Presentation Modifications?**

Reviewer #1: (No Response)

Reviewer #2: (No Response)

Reviewer #3: (No Response)

**Summary and General Comments**

Reviewer #1: 1. The main motive for this study, as stated in the introduction and elsewhere, is to avoid hemolytic complications from primaquine or tafenoquine. The assessment of G6PDd in males is very easy: it does not require either a spectrophotometric assay or BIOSENSOR, as it can be done by the fluorescent spot test or by other inexpensive screening tests. The need for a quantitative tests regards only females. Surprisingly, this is not stated anywhere. 

2. In view of the above, in order to assess BiOSENSOR the focus should be on the ‘intermediate’ group: specifically, on the success rate of the device in detecting heterozygotes with a range of G6PD activities. Unfortunately, by using a single intermediate control the work was not designed to assess this success rate. In addition, it is precisely in the intermediate group that problems were encountered.

3. The Authors should briefly explain how the BIOSENSOR works. Is G6PD measured by NADPH production, or by formazan production, or in what other way? Is Hb measured by cyanmethemoglobin production or in what other way?

4. Lyophilization is not a standard way to process blood samples for a G6PD assay (and it may be a factor in some of the problems encountered). Given the design of their study, the Authors should show what was the agreement between G6PD values obtained in one lab on a set of samples, fresh versus after lyophilisation and re-constitution; and on 0 to 48 hours after re-constitution. 

5. In heterozygous (intermediate) females what matters from the point of view of hemolysis is, rather than the level of G6PD activity in a hemolysate, the proportion of G6PDd red cells within their red cell mosaicism. Of course the two do correlate, but somewhere in the paper this important fact should be mentioned.

Reviewer #2: This study evaluated the repeatability and reproducibility of the quantitative G6PD device (SD Biosensor) using standard commercial controls with high, intermediate and low G6PD activities. The results showed high repeatability and reproducibility of the device in a multi-lab setting. However, the study showed significant variations of the results from spectrometry-based measurements, which the authors speculated may have resulted from the variations from Hemocue measurement of the hemoglobin (Hb) level. As the authors discussed, the major caveat is that both the SD Biosensor and Hemocue are designed for using fresh blood, whereas the study used lyophilized controls.

1. While the study design using the same lots of controls allows direct comparison of the performance of the SD Biosensor in different labs, all controls were non-anemic and had a similar level of Hb. In endemic conditions, many vivax patients are anemic with much lower levels of Hb. Given that the variations in results may derive from the Hb measurement, it would be appropriate if the measurement was also done using diluted controls to mimic anemic situation. 

2. Comparison between the SB biosensor and spectrometry results showed that the latter could differentiate the low from intermediate levels of G6PD activities, whereas SD biosensor could not. It is doubted whether the discrepancy indeed reflects the lower sensitivity of SD biosensor in discriminating low vs intermediate level activity or the standard controls tested in this study are not appropriate. 

Minor comments

1. Need to synchronize spellings (Br or Am, but not both) – e.g., line 37, haemolysis; line 81, hemolysis; line 43, lyophilised; line 54, lyophilized

2. Line 214-217: please clarify whether the comparison was done for Low, Intermidiate or High separately, or these reflect the combined results.

3. Figure 2: specify the shared areas are recommended….. for the Biosensor device. Also clarify what do the dots represent (individual measurement)?

4. Line 223-224. Why do different dotted lines represent the same? I think red line shows intermediate controls, while green dotted line shows the high control. The same mistakes are also found in Fig 5 and Fig 6 legends.

5. Line 236: It shows that the normalized activity was positively correlated. Are they significantly different (for Fig 1, Intermediate, it looks like the SD biosensor readings and spectrometry results are quite different).

6. For the supplementary tables, please include the statistical results in these tables too.

Reviewer #3: The authors present G6PD enzyme activity measurement comparisons by two different methods in their paper titled: The STANDARD G6PD test (SD Biosensor) shows good repeatability and reproducibility in a multi-laboratory comparison. The paper is very well written and the data analysis and presentation is clear. 

Please find below some minor comments I would like to make:

1. Would the authors please elaborate in more detail on the differences in between the recommended range set by the manufacturer and the range found by Biosensor/spectrometry. How do the company determine G6PD activity ranges? Do the company give any more information with regards to the controls i.e. from male or female, SNPs in question, country of origin of controls?

2. Would the authors please discuss in more detail the inability of Biosensor to distinguish in between low and intermediate G6PD activity, especially in phase B. It seems that the reason why there is so much overlap in phase B, is that the median low G6PD activity is higher and the median intermediate G6PD activity is similar to phase A. How, if at all, would this be influenced by different batches?

3. Would the authors please discuss the potential relevance of the inability to distinguish in between low and intermediate G6PD activity? What will it mean in terms of giving PQ to people/deciding on cut-offs/dosage regimes etc.?

The authors only describe data acquisition on controls and in well-equipped laboratory settings. It will be highly interesting to see results from field studies using the Biosensor.

PLOS authors have the option to publish the peer review history of their article (what does this mean?). If published, this will include your full peer review and any attached files.

Reviewer #1: No

Reviewer #2: No

Reviewer #3: Yes: Lynn Grignard
---

## [Decision Letter · Decision Letter 1]

25 Oct 2021

Dear Dr Ley,

Thank you very much for submitting your manuscript "The STANDARD G6PD test (SD Biosensor) shows good repeatability and reproducibility in a multi-laboratory comparison" for consideration at PLOS Neglected Tropical Diseases. As with all papers reviewed by the journal, your manuscript was reviewed by members of the editorial board and by several independent reviewers. In light of the reviews (below this email), we would like to invite the resubmission of a significantly-revised version that takes into account the reviewers' comments. 

We cannot make any decision about publication until we have seen the revised manuscript and your response to the reviewers' comments. Your revised manuscript is also likely to be sent to reviewers for further evaluation.

Sincerely,

J. Kevin Baird

Guest Editor

Mary Lopez-Perez

Deputy Editor

Reviewer's Responses to Questions

**Key Review Criteria Required for Acceptance?**

**Methods**

-Are the objectives of the study clearly articulated with a clear testable hypothesis stated?

-Is the study design appropriate to address the stated objectives?

-Is the population clearly described and appropriate for the hypothesis being tested?

-Is the sample size sufficient to ensure adequate power to address the hypothesis being tested?

-Were correct statistical analysis used to support conclusions?

-Are there concerns about ethical or regulatory requirements being met?

Reviewer #1: As I stated before, the comparison among several distant laboratory must have posed considerable challenges and it is an enterprise for which the Authors are to be commended.

Reviewer #2: (No Response)

Reviewer #3: Yes

**Results**

-Does the analysis presented match the analysis plan?

-Are the results clearly and completely presented?

-Are the figures (Tables, Images) of sufficient quality for clarity?

Reviewer #1: The Authors have responded to all queries, but with respect to some important ones I find their response not satisfactory.

General criticism 1. The Authors refuse to make the simple statement that the diagnosis of G6PD deficiency in males is easy and does not require a Biosensor. They prefer to say “especially females”.

General criticism 2. I understand that the purpose of the study was precision, because on accuracy ‘there are already published data’. To me it is a reason of concern that, with accuracy already established, it is not confirmed in this work. It does not make sense to me that the need for a quantitative test is at the basis of promoting Biosensor, and then the Authors do not know the original (true) values of G6PD activity in the samples they have distributed: I find this unacceptable.

General criticism 3. This clarification is appropriate: I think it is a good addition to the manuscript. 

General criticism 4. The Authors essentially seem to agree that lyophilization was not a good idea (as pointed out also by another Reviewer). In principle, one might question whether the repeatability assessed on lyophilized samples would be valid for fresh samples. This is buttressed by the Authors’ own statement that “Determining accuracy of Biosensor Hb readings against the Hemocue reference assay with reconstituted lyophilized controls is of limited clinical relevance.” This puts into question the entire Hb data, that are half of the data in the paper. 

General criticism 5. The added sentence is not well phrased. The G6PD activity measured in a hemolysate depends on the ratio between the two cell populations; the potential severity of hemolysis depends on the proportion of G6PD deficient red cells, because these are the ones susceptible to hemolysis from primaquine or tafenoquine.

Additional comments.

I was interested and not surprised that the “ACS recommend that laboratories develop their own in-house ranges”. This has either not been done, or the data are not shown: in my view this should be mentioned in the discussion as a limitation. In addition, I still think that readers may be confused by the fact that activity of normal samples is called “high”. As for intermediate activity, I can only reiterate that since this is the important range, it is unfortunate that only one set of samples was used within this range. 

The answer to my criticism of Fig. 5 is unsatisfactory. The Authors now say that “Clinical data will be important to further investigate the poor discriminatory power of the Biosensor at low G6PD activities“: how on earth will “clinical data” help, when the device has failed in highly qualified labs?

Reviewer #2: (No Response)

Reviewer #3: Yes

**Conclusions**

-Are the conclusions supported by the data presented?

-Are the limitations of analysis clearly described?

-Do the authors discuss how these data can be helpful to advance our understanding of the topic under study?

-Is public health relevance addressed?

Reviewer #1: Considering all of the above, and the many limitations of this study, some of which are admitted by the Authors, I find the claim of ‘consistent and robust performance’ (line 428) seriously inconsistent.

Reviewer #2: (No Response)

Reviewer #3: Yes

**Editorial and Data Presentation Modifications?**

Reviewer #1: (No Response)

Reviewer #2: (No Response)

Reviewer #3: N/A

**Summary and General Comments**

Reviewer #1: In summary, there are many unsatisfactory results in the performance of Biosensor in this study, with respect to both Hb and G6PD activity. The whole point of a quantitative test versus a qualitative test ought to be to detect heterozygotes at risk, and the device has proven wanting precisely in this respect. The Authors correctly recognize that Biosensor is inferior to the standard spectrophotometric assay. None of these serious problems is reflected in the Abstract, that gives instead the impression that everything is OK. The abstract should state instead that there are problems with the Biosensor and that the device needs to be improved.

Reviewer #2: The manuscript has been satisfactorily revised to address all my comments and suggestions.

Reviewer #3: The authors have addressed all the reviewers' comments and made changes to the manuscript accordingly.

PLOS authors have the option to publish the peer review history of their article (what does this mean?). If published, this will include your full peer review and any attached files.

Reviewer #1: No

Reviewer #2: Yes: Liwang Cui

Reviewer #3: No
---

## [Editor Report · Decision Letter 2]

10 Dec 2021

Dear Dr Ley,

Thank you very much for submitting your manuscript "The STANDARD G6PD test (SD Biosensor) shows good repeatability and reproducibility in a multi-laboratory comparison" for consideration at PLOS Neglected Tropical Diseases. As with all papers reviewed by the journal, your manuscript was reviewed by members of the editorial board and by several independent reviewers. The reviewers appreciated the attention to an important topic. Based on the reviews, we are likely to accept this manuscript for publication, providing that you modify the manuscript according to the review recommendations. 

Reviewer #1 of the original and revised submissions recommended rejection of the first revision and has declined to review the second revision. As the serving guest editor of this manuscript, I am therefore forced to weigh the merits of Reviewer #1’s strong criticisms and those of the responses and revisions of the authors. After careful study of these factors, along with a read of the second revision (unseen by Reviewer #1, henceforth “Reviewer”), I would be prepared to accept this manuscript with the following recommendations: 

1. The title should be revised to describe the study theme and design. It is not necessary to declare that which may arguably mislead one to perceive proven suitability for purpose. This study does not do that. Something like; “Repeatability and reproducibility of a portable quantitative G6PD test device within and among users and laboratories” 

2. In the abstract, line 60, the authors seem to infer inferiority of the spectrophotometric standard compared to the POC device. This is unnecessary. Simply express that good repeatability and inter-laboratory reproducibility occurred, but the device did not reliably discriminate low and intermediate samples. The argument for set global cut-off values guiding 8-aminoquinoline treatment decisions (line 78 & elsewhere) may not be supported by observations limited to relatively very few specimens, none of which represented fresh peripheral blood. 

3. If the technicians involved were not trained and certified as competent/capable in the spectrophotometric assay (as they were with the Biosensor), this should perhaps be expressed. If that is so, you may wish to express less firmly that statement at line 374-6.

4. The authors have failed to make an important point (in my opinion). The intended use of the instrument is to generate a number that informs a “go” vs. “no go” decision on 8-aminoquinoline administration. That number is 70-80% of normal G6PD activity (GSK or WHO), below which fall all “intermediate” and “deficient” phenotypes. The lack of discernment of “low” and “intermediate” by the Biosensor is thus of no practical consequence. You see my opinion on this diametrically opposes that of the Reviewer. Separating “low” and “intermediate” is certainly critical for scientific study of the phenomenon and solving the clinical problem, but in the hands of the intended end-user there is no need to separate them – both safely get the “no go” on treatment. It is a single number, greater than, or less than – the only precision of importance is in that classification. 

5. Your concluding statement (line 438-40) exceeds the limitations imposed by your experimental approach. Your work convincingly demonstrates reproducibility and repeatability of the Biosensor. Your work certainly does not prove the instrument “offers reliable quantitative diagnosis of G6PD status” – how could you possibly prove that with this design? Your evidence, in fact, seems to argue the opposite conclusion. You may articulate that the findings are consistent with an expectation that the instrument may consistently inform a safe treatment decision based on locally relevant thresholds for that decision.

Sincerely,

J. Kevin Baird

Guest Editor

Mary Lopez-Perez

Deputy Editor

Reviewer #1 of the original and revised submissions recommended rejection of the first revision and has declined to review the second revision. As the serving guest editor of this manuscript, I am therefore forced to weigh the merits of Reviewer #1’s strong criticisms and those of the responses and revisions of the authors. After careful study of these factors, along with a read of the second revision (unseen by Reviewer #1, henceforth “Reviewer”), I would be prepared to accept this manuscript with the following recommendations: 

1. The title should be revised to describe the study theme and design. It is not necessary to declare that which may arguably mislead one to perceive proven suitability for purpose. This study does not do that. Something like; “Repeatability and reproducibility of a portable quantitative G6PD test device within and among users and laboratories” 

2. In the abstract, line 60, the authors seem to infer inferiority of the spectrophotometric standard compared to the POC device. This is unnecessary. Simply express that good repeatability and inter-laboratory reproducibility occurred, but the device did not reliably discriminate low and intermediate samples. The argument for set global cut-off values guiding 8-aminoquinoline treatment decisions (line 78 & elsewhere) may not be supported by observations limited to relatively very few specimens, none of which represented fresh peripheral blood. 

3. If the technicians involved were not trained and certified as competent/capable in the spectrophotometric assay (as they were with the Biosensor), this should perhaps be expressed. If that is so, you may wish to express less firmly that statement at line 374-6.

4. The authors have failed to make an important point (in my opinion). The intended use of the instrument is to generate a number that informs a “go” vs. “no go” decision on 8-aminoquinoline administration. That number is 70-80% of normal G6PD activity (GSK or WHO), below which fall all “intermediate” and “deficient” phenotypes. The lack of discernment of “low” and “intermediate” by the Biosensor is thus of no practical consequence. You see my opinion on this is diametrically opposes that of the Reviewer. Separating “low” and “intermediate” is certainly critical for scientific study of the phenomenon and solving the clinical problem, but in the hands of the intended end-user there is no need to separate them – both safely get the “no go” on treatment. It is a single number, greater than, or less than – the only precision of importance is in that classification. 

5. Your concluding statement (line 438-40) exceeds the limitations imposed by your experimental approach. Your work convincingly demonstrates reproducibility and repeatability of the Biosensor. Your work certainly does not prove the instrument “offers reliable quantitative diagnosis of G6PD status” – how could you possibly prove that with this design? Your evidence, in fact, seems to argue the opposite conclusion. You may articulate that the findings are consistent with an expectation that the instrument may consistently inform a safe treatment decision based on locally relevant thresholds for that decision.

Figure Files:

Data Requirements:

Reproducibility:

References

---

## [Editor Report · Decision Letter 3]

17 Jan 2022

Dear Dr Ley,

We are pleased to inform you that your manuscript 'Repeatability and reproducibility of a handheld quantitative G6PD diagnostic' has been provisionally accepted for publication in PLOS Neglected Tropical Diseases.

Best regards,

J. Kevin Baird

Guest Editor

Mary Lopez-Perez

Deputy Editor

---

## [Editor Report · Acceptance letter]

26 Jan 2022

Dear Dr Ley,

We are delighted to inform you that your manuscript, " Repeatability and reproducibility of a handheld quantitative G6PD diagnostic ," has been formally accepted for publication in PLOS Neglected Tropical Diseases.

Best regards,

Shaden Kamhawi

co-Editor-in-Chief

Paul Brindley

co-Editor-in-Chief
